# A randomized clinical trial assessing the effect of automated medication-targeted alerts on acute kidney injury outcomes

F. Perry Wilson [1] ✉, Yu Yamamoto [1], Melissa Martin[1],
Claudia Coronel-Moreno[1,2], Fan Li[3], Chao Cheng[3], Abinet Aklilu[1], Lama Ghazi[1,4],
Jason H. Greenberg[1], Stephen Latham[5], Hannah Melchinger [1],
Sherry G. Mansour[1], Dennis G. Moledina [1], Chirag R. Parikh [6],
Caitlin Partridge[2], Jeffrey M. Testani[7] & Ugochukwu Ugwuowo[1]

Acute kidney injury is common among hospitalized individuals, particularly those exposed to certain medications, and is associated with substantial morbidity and mortality. In a pragmatic, open-label, National Institutes of Health-funded, parallel group randomized controlled trial (clinicaltrials.gov NCT02771977), we investigate whether an automated clinical decision support system affects discontinuation rates of potentially nephrotoxic medications and improves outcomes in patients with AKI. Participants included 5060 hospitalized adults with AKI and an active order for any of three classes of medications of interest: non-steroidal anti-inflammatory drugs, renin-angiotensin-aldosterone system inhibitors, or proton pump inhibitors. Within 24 hours of randomization, a medication of interest was discontinued in 61.1% of the alert group versus 55.9% of the usual care group (relative risk 1.08, 1.04 – 1.14, p = 0.0003). The primary outcome – a composite of progression of acute kidney injury, dialysis, or death within 14 days - occurred in 585 (23.1%) of individuals in the alert group and 639 (25.3%) of patients in the usual care group (RR 0.92, 0.83 – 1.01, p = 0.09). Trial Registration Clinicaltrials.gov NCT02771977.

Acute kidney injury (AKI)—an abrupt decline in kidney function—is a major cause of morbidity and mortality among hospitalized patients, associated with an increased hospital length of stay and a 10-fold increase in inpatient mortality[1–3]. AKI also carries an independent risk for development of chronic kidney disease (CKD), end stage kidney disease, and death[4–7].

Despite current international best practice guidelines for management of AKI focusing on avoidance of nephrotoxic exposures, studies have shown that AKI, often asymptomatic in presentation, frequently goes unnoticed, and appropriate workup and treatment is inconsistently performed[8–12]. For example, a retrospective study of 9534 hospitalized patients with severe AKI found that the rate of discontinuation of potentially nephrotoxic medications is low in the early stages of AKI, revealing an opportunity for early intervention that could potentially prevent progression to more severe forms of AKI[13].

[1]Clinical and Translational Research Accelerator. Department of Medicine, Yale School of Medicine, New Haven, CT, USA. [2]Joint Data Analytics Team. Yale New Haven Health System, New Haven, CT, USA. [3]Department of Biostatistics, Yale School of Public Health, New Haven, CT, USA. [4]Department of Epidemiology, University of Alabama School of Public Health, Birmingham, Alabama, USA. [5]Interdisciplinary Center for Bioethics, Yale University, New Haven, CT, USA. [6]Division of Nephrology. Department of Medicine, Johns Hopkins School of Medicine, Baltimore, MD, USA. [7]Section of Cardiovascular Medicine. Department of Medicine, Yale School of Medicine, New Haven, CT, USA. ✉e-mail: Francis.p.wilson@yale.edu

Medications that affect kidney function are common contributors to AKI in hospitalized patients. However, international guidelines vary with respect to discontinuation of non-steroidal anti-inflammatory drugs (NSAIDs), renin-angiotensin-aldosterone system inhibitors (RAASi), and proton pump inhibitors (PPIs). Observational studies have shown that NSAIDs, potentially by increasing kidney vasoconstriction, increase the risk of AKI both in and out of the hospital[14–16]. International guidelines strongly recommend the discontinuation of NSAIDs in the context of AKI[8,9]. RAASi lower hydrostatic pressure at the glomerulus, thus decreasing the glomerular filtration rate and inhibiting the clearance of uremic toxins[17,18]. Interestingly, RAASi may also increase kidney peritubular blood flow, potentially protecting against ischemic damage[19,20]. There is thus debate regarding the utility of RAASi discontinuation during AKI, and this is reflected in guideline recommendations to "consider" discontinuation[21]. PPIs have long been associated with the development of acute interstitial nephritis (AIN), a form of AKI, and have been linked to the progression of CKD[22–25]. As AIN is likely underdiagnosed in hospitalized patients, and PPIs are likely overprescribed, there is significant interest in whether empiric discontinuation of PPI would affect the course of AKI in this population[26–29]. To date, international guidelines do not address empiric discontinuation of PPI among patients with AKI. The **EL**ectronic **A**lerts for AK**I A**melioration 2 (ELAIA-2) investigators set out to evaluate the aggregate and individual effect of prompting discontinuation of NSAID, RAASi, and PPI within a single trial framework to provide higher quality evidence to clinicians caring for patients with AKI exposed to these agents.

Clinical decision support and best practice alerts have proliferated with the adoption of the electronic health record (EHR), often without rigorous evidence-based support. Trials of alerts for patients with AKI, including our own, have led to mixed results, though the most successful efforts couple alerts to specific, actionable information[30–33]. In the ELAIA-2 trial, we set out to evaluate the effectiveness of a medication-targeted AKI alert across multiple hospitals within a large health system. Here, we show that alerts can increase the rate of cessation of medications of interest, but have limited effect on key clinical outcomes.

### Role of the funding source
The funder of the study had no role in study design, data collection, data analysis, data interpretation, or writing of the report.

## Results
From August 16, 2020 to November 29, 2021, 5060 participants were enrolled. The study population was consistent with a hospitalized AKI population (Table 1). The median (IQR) age was 70 (59–80) years, 2453 (48%) were women, and 968 (19%) described themselves as Black. Common comorbidities included hypertension (68%), congestive heart failure (32%), and CKD (25%).

At the time of randomization, 1553 (31%) of patients were receiving NSAIDs, 2679 (53%) were receiving RAASi and 3298 (65%) were receiving PPIs. Overlap of MOIs was common (Supplemental Fig. 1), with 2139 (42%) of patients receiving more than one MOI.

### Effect of alerts on progression of AKI, dialysis, and death
On the basis of a prespecified interim analysis (N = 1980), which found that the primary outcome occurred in 255 (24.7%) of individuals in the alert group and 254 (26.8%) of those in the usual care group (p = 0.30), the external DSMB recommended the trial proceed to full recruitment. In the final analysis (N = 5060), the primary outcome occurred in 585 (23.1%) of individuals in the alert group and 639 (25.3%) of patients in the usual care group (RR 0.92, 0.83–1.01, p = 0.09). The effect of the alert among pre-specified subgroups defined by exposure to a given MOI demonstrated a benefit among those exposed to PPIs. In the PPI-exposed subgroup (n = 3298), 445 (27%) of individuals in the alert arm

**Table 1 | Baseline characteristics**

| Characteristic | Alert (N = 2532) | Usual care (N = 2528) |
|---|---|---|
| **Demographics** | | |
| Age (years) | 70 (59,81) | 70 (59,80) |
| Female sex | 1231 (49%) | 1222 (48%) |
| Black | 498 (20%) | 470 (19%) |
| Hispanic | 350 (14%) | 341 (13%) |
| **Hospital location** | | |
| Medical admission | 1937 (77%) | 1924 (76%) |
| Patient in the ICU | 560 (22%) | 598 (24%) |
| Patient in the emergency department | 90 (4%) | 87 (3%) |
| Patient in the ward | 1775 (70%) | 1746 (69%) |
| Hospital 1 | 1197 (47%) | 1250 (49%) |
| Hospital 2 | 623 (25%) | 591 (23%) |
| Hospital 3 | 464 (18%) | 446 (18%) |
| Hospital 4 | 248 (10%) | 241 (10%) |
| **Comorbidities** | | |
| Chronic kidney disease | 671 (27%) | 616 (24%) |
| Congestive heart failure | 827 (33%) | 784 (31%) |
| COPD | 776 (31%) | 762 (30%) |
| Diabetes mellitus | 967 (38%) | 928 (37%) |
| Hypertension | 1710 (68%) | 1726 (68%) |
| Malignancy | 549 (22%) | 563 (22%) |
| Depression | 601 (24%) | 580 (23%) |
| Liver disease | 310 (12%) | 359 (14%) |
| **AKI stage at randomization** | | |
| Stage 0[a] | 6 (0.24%) | 3 (0.12%) |
| Stage 1 | 2279 (90%) | 2248 (89%) |
| Stage 2 | 191 (7.5%) | 230 (9.1%) |
| Stage 3 | 56 (2.2%) | 47 (1.9%) |
| **Laboratory values** | | |
| eGFR on admission (ml/min/1.73 m$^2$) | 60 (38, 87) | 61 (38, 88) |
| Creatinine at randomization (mg/dL) | 1.5 (1.2,2) | 1.5 (1.1,2) |
| Creatinine at admission (mg/dL) | 1.2 (0.9,1.7) | 1.2 (0.8,1.6) |
| Sodium (meq/L) | 138 (135,141) | 138 (135,141) |
| Potassium (meq/L) | 4.2 (3.8,4.6) | 4.2 (3.8,4.6) |
| Chloride (meq/L) | 102 (99,106) | 102 (99,106) |
| Bicarbonate (meq/L) | 23 (21,26) | 23 (20,26) |
| Anion gap (meq/L) | 12 (10,14) | 12 (10,14) |
| Blood urea nitrogen (mg/dL) | 29 (20,42) | 29 (19,42) |
| White blood cell count (×1000/µL) | 9.5 (6.8,13.5) | 9.6 (6.7,13.6) |
| Hemoglobin (g/dL) | 10.6 (8.8,12.5) | 10.6 (8.9,12.4) |
| Platelet count (×1000/µL) | 213 (153,287) | 214 (149,288) |
| Modified SOFA Score | 2 (1, 4) | 3 (1, 5) |
| **Exposures prior to AKI** | | |
| Contrast in prior 72 h | 621 (25%) | 662 (26%) |
| RAASi | 1350 (53%) | 1329 (53%) |
| NSAID | 748 (30%) | 805 (32%) |
| PPI | 1654 (65%) | 1644 (65%) |
| One medication of Interest | 1470 (58%) | 1451 (57%) |
| Two medications of interest | 904 (36%) | 904 (36%) |
| Three medications of interest | 158 (6%) | 173 (7%) |

Data is presented as median (interquartile range) or count (percentage).

*ICU* intensive care unit, *COPD* chronic obstructive pulmonary disease, *eGFR* estimated glomerular filtration rate, *SOFA* sequential organ failure assessment, *AKI* acute kidney injury, *RAASi* renin-angiotensin-aldosterone system inhibitor, *NSAID* non-steroidal anti-inflammatory drug, *PPI* proton pump inhibitor.

[a]Stage 0 AKI occurs because patients are randomized and, subsequently, the creatinine that led to randomization is revised lower (to below the AKI threshold) due to lab error.

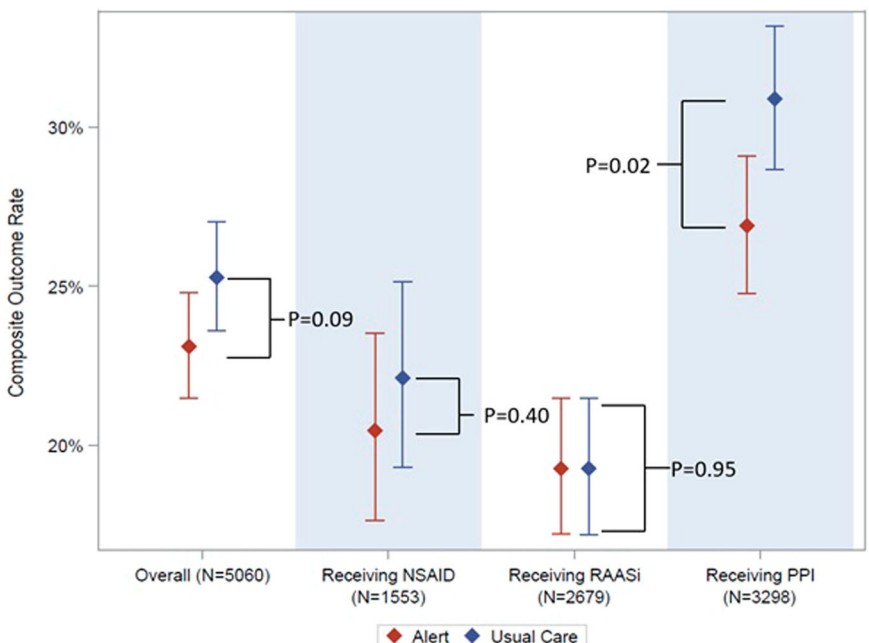

**Fig. 1 | Composite outcome rates.** Dot represents proportion. Bars represent 95% confidence intervals. NSAID: Non-steroidal anti-inflammatory drug. RAASi renin-angiotensin-aldosterone-system inhibitor, PPI proton pump inhibitor. *P*-values from Cochran-Mantel-Haenszel chi-square tests. *P*-value for RAASi: $8.55 \times 10^{-7}$. Source data are provided as a Source Data file.

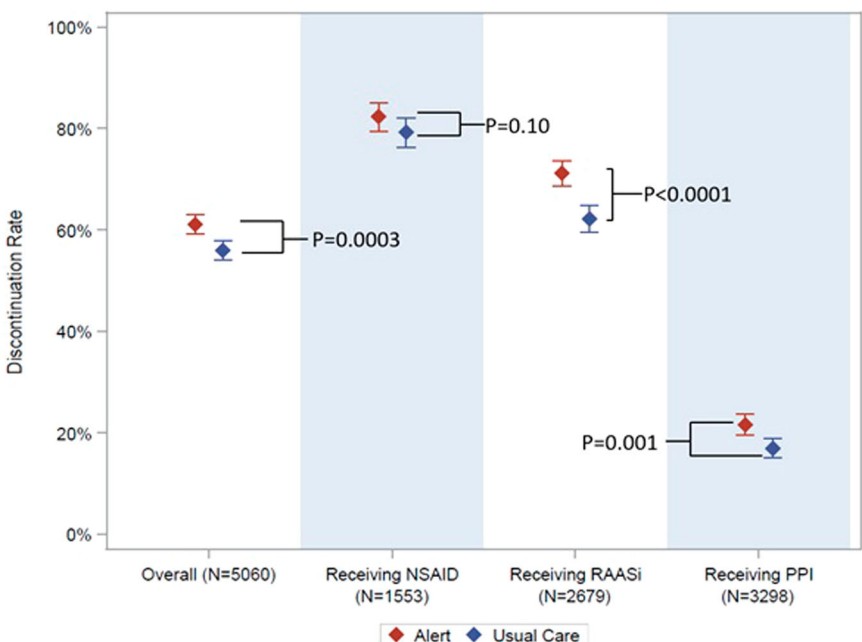

**Fig. 2 | Medication of interest discontinuation rate.** Dot represents proportion. Bars represent 95% confidence intervals. NSAID: Non-steroidal anti-inflammatory drug. RAASi renin-angiotensin-aldosterone-system inhibitor, PPI proton pump inhibitor. *P*-values from Cochran-Mantel-Haenszel chi-square tests. Source data are provided as a Source Data file.

experienced the primary outcome compared to 508 (31%) of those in the usual care arm (RR 0.88, 0.79–0.98, *p* = 0.02) (Fig. 1). Baseline characteristics of MOI-exposed subgroups appear in Supplemental Tables 4, 5, 6.

### Mediation analysis of alert effects
Alerts may provide patient benefit beyond medication discontinuation (such as by promoting further diagnostic workup, treatment, or avoidance of nephrotoxins). As the PPI subgroup was the only one to show a significant overall effect of the alert, we performed mediation analysis to determine if the clinical effect of the alert was driven by cessation of PPI. In the unadjusted analysis without any baseline covariates (Supplemental Table 7), we observed that the alerts decreased the relative odds of death, dialysis, and progression of AKI by 18% (total effect on odds ratios of 0.82; 95% CI: [0.72, 0.97]), and that 10.7% of that total effect was mediated through PPI cessation (95% CI: [2.9%,

**Table 2 | Secondary outcomes**

| Outcome | Alert (N = 2532) | Usual care (N = 2528) | Relative risk (95% CI) |
|---|---|---|---|
| Progression of AKI | 475 (18.8%) | 505 (20.0%) | 0.95 (0.85–1.06) |
| Dialysis | 123 (4.9%) | 127 (5.0%) | 0.98 (0.77–1.25) |
| Death | 253 (10.0%) | 282 (11.2%) | 0.90 (0.77–1.06) |
| Inpatient mortality | 356 (14.1%) | 365 (14.4%) | 0.98 (0.86–1.12) |
| Inpatient dialysis | 151 (6.0%) | 137 (5.4%) | 1.12 (0.89–1.4) |
| Progression to stage 2 AKI | 242 (9.6%) | 248 (9.8%) | 0.98 (0.83–1.16) |
| Progression to stage 3 AKI | 231 (9.1%) | 256 (10.1%) | 0.91 (0.77–1.08) |
| 30-day readmission | 322 (12.7%) | 354 (14.0%) | 0.91 (0.79–1.05) |
| Inpatient kidney consult | 367 (14.5%) | 366 (14.5%) | 1.01 (0.88–1.15) |
| Duration of AKI (median days, IQR) | 1 (0.8, 2.1) | 1.1 (0.8, 2.2) | – |
| Length of stay (post randomization) (median days, IQR) | 5.3 (2.3–11.8) | 5.2 (2.2–11.2) | – |
| Cost of Hospitalization, $ | 26,900 (12,400–63,600) | 27,900 (12,500–63,400) | – |

Unless otherwise specified, all outcomes are evaluated within 14 days of randomization or hospital discharge and expressed as count, percentage.
*AKI* acute kidney injury. Difference in duration of AKI, LOS, and total hospital costs were assessed via the VanElteren test (accounting for clustering within hospitals), and were not significantly different [duration of AKI ($p = 0.14$), length of stay ($p = 0.38$), cost of hospitalization ($p = 0.66$)].

44.7%]). We repeated the analysis adjusting for potential confounders, and the results remained similar.

### Key secondary analyses

Within 24 h of randomization, an MOI was discontinued in 61.1% of the alert group and 55.9% of the usual care group (relative risk 1.08, 1.04–1.14, $p = 0.0003$). The NSAID was discontinued in 82% of the alert group and 79% of the usual care group (RR 1.04, 0.99–1.09). RAASi were discontinued in 71% of the alert group and 62% of the usual care group (RR 1.14, 1.08–1.21). PPIs were discontinued in 22% of the alert group and 17% of the usual care group (RR 1.26, 1.10–1.45) (Fig. 2).

Secondary outcomes appear in Table 2. There was no significant effect of alerting on the individual components of the composite outcome, nor did we observe an effect on duration of AKI, 30-day readmission, the rate of kidney consults, or total hospital cost. Among the PPI group (Supplemental Table 8), alerting was associated with a 12% reduced risk of progression of AKI ($p = 0.05$). Manual adjudication of AKI recognition is ongoing at the time of this publication and will be reported in a future manuscript.

### Subgroup analyses

Figure 3 demonstrates the effect of the alert on the composite outcome across pre-specified subgroups of interest. There was no significant change in alert effect across any clinical subgroup, including among patients in the ICU vs. Non-ICU, those admitted to a surgical versus a medical service, or based upon the baseline serum creatinine concentration. There was significant heterogeneity of alert effect across the four study hospitals ($p = 0.03$).

### Alert safety

Pre-specified safety outcomes appear in Table 3. In the NSAID group, we did not observe a significant increase in the use of opioids or higher pain scores in the alert group. In the RAASi group, we did not observe significant differences in blood pressure parameters or the need for mechanical ventilation. In the PPI group, we did not observe differences in the rate of blood transfusion, the minimum hemoglobin level attained, or the maximum pain score.

### Assessment of contamination

In theory, providers could "learn" to discontinue MOIs when they receive alerts, and apply that knowledge to patients randomized to usual care. To assess the extent of contamination across study arms, we conducted several analyses. First, we examined whether the effect of the alert was attenuated the longer the alert had been active in a study hospital and did not find a significant effect (interaction $p = 0.51$, Supplemental Fig. 2A/B). Second, we examined the effect of prior provider exposure to these alerts on alert efficacy. In this analysis, we found no significant interaction between the number of alerts a provider had seen prior to a given alert and the overall efficacy of alerting ($p$-for-interaction = 0.48). While we had planned to compare historical outcome rates, these are less informative than we had hoped, as this trial was conducted during the COVID pandemic, which seems to have been associated with an increased acuity of illness in this population (historical composite outcome rate for theoretically eligible patients (N = 1074) 16.8% versus 26.8% in the control group of this study).

## Discussion

This pragmatic, randomized trial of medication-targeted electronic alerts for AKI found that alerts had a significant impact on reducing exposure to the medications of interest, but no significant effect on the primary outcome of progression of AKI, dialysis, or death. However, the alert was associated with clinical benefit among individuals receiving PPI at the time of randomization; PPI-exposed individuals comprised the majority of the study cohort.

Prior research into the efficacy of alerting for AKI has shown mixed results. In part, this is due to the variety of study designs employed to evaluate AKI alerts (which range from retrospective before/after analyses, to stepped-wedge designs, to individual-level randomized trials)[34]. In addition, the nature of the alerts studied varies significantly, ranging from e-mail based notifications to electronic "pop ups" of the sort tested in this study. Our group has conducted two prior randomized trials evaluating purely informational AKI alerts which, while showing a modest effect on process outcomes, did not demonstrate an improvement in clinical outcomes[30, 31].

Several studies show that AKI alerts have substantial promise when executed appropriately. A stepped-wedge trial by Selby et al. showed that AKI alerts coupled to a care bundle and an educational AKI package led to a significantly reduced hospital length of stay, but no change in overall mortality[32]. A large before / after study of a clinical decision support system for AKI including more than 500,000 patients found that hospital mortality decreased by around 10% among those with AKI compared to no change in those without AKI between the pre- and post-intervention period, but this study design precluded a causal analysis[35]. The Nephrotoxic Injury Negated by Just-in-Time Action (NINJA) program has successfully demonstrated how alerts targeting medications can improve outcomes in children with AKI[33]. Together with those studies, our study suggests that alerts coupled with

**Fig. 3 | Subgroup analyses.** Bands reflect 95% confidence intervals. ICU Intensive care unit. *P*-values reflect a subgroup-by-alert interaction in a binomial regression model adjusted for hospital, except for the *p*-values for hospital, which reflect the Breslow-Day test for homogeneity. Source data are provided as a Source Data file.

actionable processes may improve outcomes for certain patients with AKI.

We were surprised to see no clinical benefit of alerts in the NSAID subgroup. We targeted this class of medications given the broad consensus and international guidelines that suggest discontinuing NSAIDs in the setting of AKI given their known potent adverse

### Table 3 | Safety outcomes

| Outcome | Alert | Usual care | Difference, 95% CI |
|---|---|---|---|
| **NSAID subgroup** | *N* = 748 | *N* = 805 | |
| Opioid prescription | 509 (68.0%) | 557 (69.2%) | 0.6 (-4–5.2) |
| Max pain score | 8 (5,10) | 8 (5,10) | 0 (−0.1, 0.1) |
| **RAASi subgroup** | *N* = 1350 | *N* = 1329 | |
| Max SBP | 162 (145,179) | 161 (145,179) | 1 (−1.2, 3.2) |
| Max DBP | 89 (80,99) | 89 (81,100) | 0.5 (−1.1, 2.1) |
| Mechanical ventilation | 181 (13.4%) | 177 (13.3%) | 0 (−2.5, 2.6) |
| **PPI subgroup** | *N* = 1654 | *N* = 1644 | |
| PRBC transfusion | 433 (26.2%) | 448 (27.3%) | 0.5 (−2.5, 3.5) |
| Minimum hemoglobin | 8.6 (7.2,10.5) | 8.6 (7.1,10.4) | 0 (−0.2, 0.2) |
| Max pain score | 7 (4,9) | 7 (3,9) | 0 (−0.5, 0.5) |

Data is count (%) or median (IQR).
*NSAID* non-steroidal anti-inflammatory drug, *RAASi* renin-angiotensin-aldosterone-system inhibitor, *PPI* proton-pump inhibitor.

biological effects on kidney health[8,9]. While the alert did numerically increase the rate of cessation of NSAIDs, the effect was not statistically significant. The control rate of NSAID cessation was the highest of any class, indicating that many physicians are stopping NSAIDs during AKI. Alerts can only translate to clinical benefit when they induce a substantial change in provider behavior.

Our observation that alerts reduced the primary outcome among those treated with PPIs has several potential explanations. Although this was a pre-specified subgroup analysis, there is the possibility that this signal is due to chance. Alternatively, there may be a causal link between PPI-use and AKI which is ameliorated by cessation in the context of alerting. Prior research, predominantly observational, has elucidated associations between PPI and AKI, and particularly acute interstitial nephritis[23,24,36,37]. Research has also suggested a link between PPI use and CKD[22,25,38]. We also recognize that the alert may have effects on AKI diagnosis, management, and treatment beyond medication cessation alone. In that context, the observed benefit among those receiving PPIs may be due to the fact that PPI use flags a distinct population of patients that benefit from AKI alerts in general. We note, for example, that those receiving PPIs were in more acutely ill than those who were not receiving PPI—with a higher proportion randomized in the ICU, and overall worse outcomes. Our mediation analysis, though underpowered, suggests that there are multiple pathways of alert benefit in this population, with some, but likely not the majority, benefit mediated by actual cessation of the PPI. Further studies to elucidate the mechanism of PPI-associated AKI and the benefit of withholding these medications in the setting of changes in kidney function are needed.

We were also intrigued to examine the data surrounding the alert targeting those on RAASi. Substantial debate exists about the utility of stopping these agents during AKI. RAASi reduce glomerular hydrostatic pressure and thus GFR which may lead to harm by decreasing clearance of uremic byproducts[39]. On the other hand, they increase kidney perfusion which theoretically may reduce ongoing ischemia and allow for more rapid AKI recovery[40]. While the alert led to significantly greater cessation of RAASi, we saw no effect on clinical outcomes, suggesting that these two forces may be in balance. Future studies may examine the benefit of stopping (or continuing) RAASi in key patient populations—such as those with AKI and heart failure.

Limitations of this study include the fact that only three medication classes were targeted. There are many other medications, including aminoglycoside antibiotics, certain types of chemotherapy, and others that are potentially nephrotoxic. However, we designed this trial not only to inform the benefit of alerting for AKI in the setting of medication exposures, but to evaluate the impact of cessation of key often-used medication classes in the setting of AKI. This has allowed us to conduct careful sub-group analyses that suggest that, for example, cessation of PPI may be reasonable among those with AKI. Other limitations include the fact that this trial was randomized at the patient level, which introduces the possibility of contamination, as providers who receive alerts may learn how better to care for AKI patients over time. Alternative designs, such as cluster-randomized or stepped-wedge designs might ameliorate this, but simulation studies have shown that, in the setting of potential contamination, statistical power is maximized through inflation of sample size (as done here) more efficiently than by conversion to a cluster or stepped-wedge design[41,42]. In addition, the lack of effect modification by study time suggests that there was limited contamination of the usual care arm of the study. This study was conducted within a single health system in the Northeastern US, potentially limiting generalizability, but the hospitals included serve a demographically and economically diverse population. In addition, our alerts did not exist in a vacuum, but in competition with all the other clinical alerts and distractions that are enabled by the EHR. Alert fatigue may diminish alert performance. Moreover, in order to meet requirements set out by the IRB, the language of the alert was limited in how stringently it could recommend medication cessation; more forceful wording could have increased cessation rates, and, potentially, clinical effectiveness. Finally, AKI is a heterogenous disease; alerts may have different efficacy depending on the underlying mechanism (for example, alerts to stop NSAIDs might be particularly useful in the setting of hemodynamic AKI). However, the ability to determine the underlying cause of AKI in real time is difficult even for trained nephrologists, more so for electronic algorithms.

Strengths of this study include its sample size, which was adequate to detect a clinically meaningful impact of alerting, its use of a clinical outcome, focus on agents commonly used in the inpatient setting, randomization, and diverse patient mix.

In conclusion, this pragmatic, randomized trial of automated, electronic alerting for individuals with AKI exposed to common kidney-relevant medications suggests that there is unlikely to be broad-based benefit to all patients, but those exposed to PPIs may benefit.

## Methods

### Study design
The protocol was carried out according to the principles in the Declaration of Helsinki and approved by the Yale Human Investigations Committee (HIC #2000025786) which approved a waiver of informed consent. A reliance agreement was obtained from the Bridgeport Hospital and Greenwich Hospital IRBs as well. A waiver of informed consent was granted on the basis that the intervention was minimal risk, that the rights and welfare of participants were protected, and that the study could not be feasibly conducted with consent (as the act of consent would serve to be an AKI alert of sorts). ELAIA-2 was a multicenter, parallel-group, pragmatic, open-label randomized trial of a real-time EHR alert for AKI among individuals exposed to one or more of three classes of kidney-relevant MOIs: NSAIDs; RAASi and PPIs (Supplemental Table 1). These three classes were chosen as consensus recommendations regarding their empiric discontinuation in patients with AKI vary. International guidelines recommend discontinuing NSAIDs in AKI[8,9]. There is substantial debate about the utility of discontinuing RAASi in AKI[21]. PPIs, while recognized to be associated with AKI, are not currently recommended to be empirically discontinued. The trial was pre-registered at https://clinicaltrials.gov/ct2/show/NCT02771977 on May 03, 2016 prior to any patient enrollment and no protocol modifications occurred after the study was launched. Patient enrollment was completed in November, 2021. The study was monitored by an external data safety and monitoring board who approved continuation after interim analysis and release of the deidentified dataset. Participants were not aware of their participation and were not compensated.

### Participants
A flow diagram of study recruitment appears as Fig. 4. Enrollment was automated, with inclusion and exclusion criteria built into the alert system itself. Eligible patients were adults >= 18 years old who were admitted to one of four participating study hospitals (Supplemental Table 2), had AKI as defined by the Kidney Disease: Improving Global Outcomes (KDIGO) serum creatinine criteria (at least 50% increase in creatinine within 7 days or a 0.3 mg/dL increase within 48 h), and had an active order for one or more of the three MOIs[9]. Patients were excluded if their initial creatinine after admission was greater than or equal to 4.0 mg/dL or if they had received dialysis within the year prior to meeting the AKI definition (as these may be patients with end-stage kidney disease cared for outside of our health system), if they had been admitted to a hospice service or had an active "comfort measures only" order, if they had a diagnosis code consistent with end-stage kidney disease, or if they had a kidney transplant within the 6 months prior to randomization (as transplant recipients at our institution receive specialist nephrology care during their inpatient stay, regardless of AKI). Patients who had an admission date prior to the "go-live" date of the alert, who triggered randomization after a discharge order had been entered or had already been randomized during a previous hospitalization were also excluded.

### Randomization and masking
Allocation was in a 1:1 ratio, achieved via simple randomization within the EHR. Participants and providers were necessarily not masked to the intervention, but the study investigators were.

### Intervention
The intervention was a "pop-up" electronic alert embedded within the EHR (Epic systems, Verona WI) (Fig. 5). The alert would display when a provider (MD, DO, PA, or APRN, regardless of whether a primary or consulting provider) opened the order entry tab of the patient's chart. Alerts would display once per provider on a 24-h basis, provided the patient still met AKI criteria and continued to have an active prescription for one of the MOIs. The alert prompted providers to "consider clinical indication for the following medications!" It included information regarding the patient's creatinine level and trend and enumerated the class of MOI and specific MOI that was currently ordered. It also provided a direct link to the order entry system to allow for potential discontinuation of the listed medication.

Patients randomized to the control group were identified via a silent alert (not shown to providers, but logged in the system), and

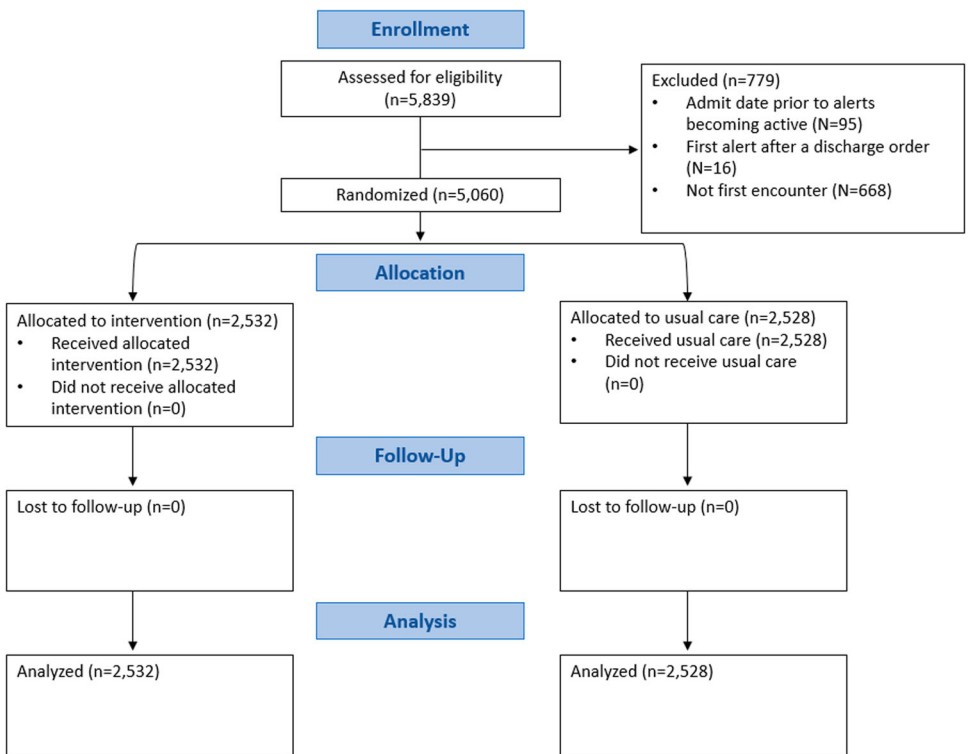

**Fig. 4 | CONSORT diagram.** Figure illustrates the flow of patients through the trial.

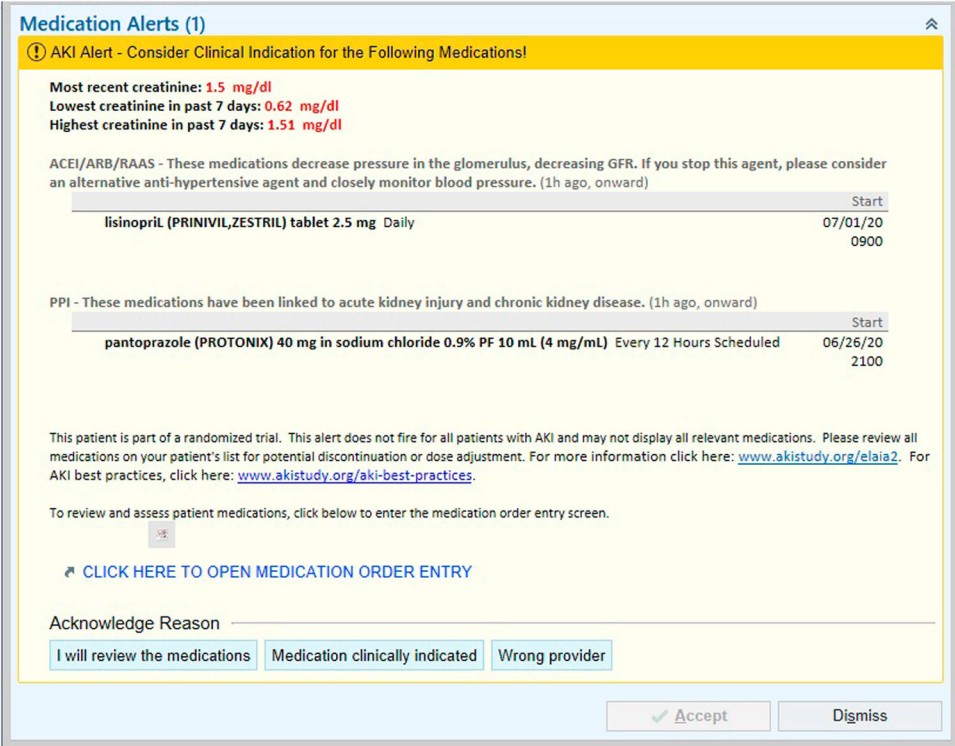

**Fig. 5 | Alert screenshot.** The alert provided information about the creatinine trend, classes of MOIs ordered and the specific MOIs currently ordered, as well as a link to the order entry screen.

received usual care. Data were collected electronically and confirmed by two chart reviewers on a random sample of alert and control patients (N = 100). All data were electronically collected through SQL queries of the Epic Electronic Health Record "Clarity" data platform,

reflect clinical care and were recorded in a secure, HIPAA compliant server.

There was no missing data on outcomes, and missing data overall was scarce (Supplemental Table 3).

## Outcomes

The primary outcome was a composite of progression of AKI, receipt of dialysis, or death within 14 days of randomization or hospital discharge. Progression of AKI was defined by achieving a higher KDIGO AKI stage than the stage present at initial randomization (stage 2 represents a doubling of creatinine, stage 3 a tripling of creatinine).

Secondary outcomes included the components of the primary outcome (at 14 days and during the entire admission), the proportion of patients who did not receive one or more MOIs for which they were enrolled within 24 h and while AKI was still present, the duration of AKI, the rate of readmission at 30-days, and hospital costs (as determined by internal cost accounting independent of insurance status). We considered AKI resolved when the most recent measured creatinine no longer met AKI criteria[43].

Strength of guidance regarding empiric discontinuation of the MOIs differed by medication class, as well as by the mechanisms by which they influence renal function, prompting us to pre-specify subgroup analyses stratified by the MOI being received at randomization. We further identified key safety outcomes that may be associated with discontinuation of each class of MOI−all within 14 days or hospital discharge. For NSAIDs, we assessed the rate of prescription of opioid analgesics and maximum pain scores. For RAASi, we assessed the maximum achieved systolic and diastolic blood pressure and the rate of intubation (a proxy for flash pulmonary edema). For PPIs, we assessed the rate of blood transfusion (as inappropriate cessation of PPI may increase the risk of gastrointestinal bleeding), the minimum hemoglobin achieved, and maximum pain scores to capture epigastric pain consequent to PPI cessation.

Based on data from our prior trial, we expected the primary outcome to occur in 18.2% of control patients and considered a relative reduction of 20−14.6% would be clinically significant. The target sample size of 5060 individuals provided 90% power to detect a difference at least this large, accounting for two interim analyses and potential 10% contamination of intervention across study arms (as providers exposed to alerts might learn to adjust their care for patients in the usual care arm).

The original protocol called for two interim analyses, one at 50% recruitment across four teaching hospitals, and one at 50% recruitment at two non-teaching hospitals, as our prior trial showed a potential signal of harm at the non-teaching hospitals. However, despite no harm signal at the initial interim analysis, the IRB did not grant our request to extend the study into the two non-teaching hospitals. We thus completed recruitment entirely in the four teaching hospitals and completed only one interim analysis.

## Statistical analysis

We present data as median (interquartile range−IQR) or count (proportion) as appropriate. We compared categorical variables across the treatment groups with the use of the Cochrane-Mantel-Haenszel Chi-squared test stratified by the four study hospitals. We assessed continuous outcomes using the VanElteren test, accounting for clustering by study hospital. All data was analyzed using the intention-to-treat principle. Subgroup analyses presented herein were pre-specified (study protocol in supplemental material). Due to the interim analysis, the two-sided $p$-value threshold for statistical significance of the primary outcome was 0.0498 based on the Hwang-Shih-DeCani spending function[44]. Secondary outcomes are evaluated at a two-sided $p$-value threshold of 0.05 without multiplicity adjustment and should be considered hypothesis generating. Subgroup analyses, while pre-specified, should be considered exploratory.

We determined whether observed clinical effects were driven by cessation of target agents by performing mediation analyses treating cessation of the MOI as a binary mediator[39]. The product of coefficients method was used to evaluate whether MOI cessation mediates the relationship between intervention and composite outcome. Baseline covariates, including age, sex, ICU at randomization, creatinine, hemoglobin, platelet counts, the use of other MOIs and time from admission to randomization, were adjusted to control for mediator-outcome confounding, and bootstrapping was used to calculate the 95% confidence interval for proportion mediated.

All statistical tests were performed in Stata version 15 (College Station, TX), SAS v9.4 (Cary, NC), and R v4.0.1 (Boston, MA).

## Patient and public involvement

Patients and the public were not involved in the design or analysis of this study.

## Reporting summary

Further information on research design is available in the Nature Portfolio Reporting Summary linked to this article.

## Data availability

The data generated in this study have been deposited in the DataDryad database under accession code https://doi.org/10.5061/dryad.kh189327p[45]. This data can be accessed and used for scientific research without permission from the authors. The DSMB has approved the release of this deidentified dataset. Source data and statistical analysis plan are provided with this paper. Source data are provided with this paper.

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

## Acknowledgements

This study was funded by the National Institute of Digestive, Diabetes, and Kidney Disease (NIDDK).

## Author contributions

F.P.W. is the guarantor and was responsible for obtaining funding, study design, data collection, data analysis, manuscript preparation, and submission. Y.Y. was responsible for the data analysis and participated in editing of the manuscript. M.M. was responsible for administrative compliance and participated in editing the manuscript. C.C.M. was responsible for obtaining data for the study. F.L. aided in the statistical design of the study and, along with C.C., participated in data analysis. A.A. aided in the protocol development and participated in editing the manuscript. L.G., J.H., S.L., H.M., S.M., D.G.M., U.U., J.M.T. and C.R.P. aided in the study conceptualization and design and participated in editing the manuscript. C.P. integrated the alert into the EHR and aided in data acquisition. The corresponding author attests that all listed authors meet authorship criteria and that no others meeting the criteria have been omitted.

## Competing interests

Competing interests: All authors have completed the ICMJE uniform disclosure form and declare: F.P.W. recognizes support from R01DK113191, R01HS027626 and P30DK079310, D.G.M. received support from National Institutes of Health awards K23DK117065 and R01DK128087, and C.R.P. is supported by NIH grants R01HL085757; J.M.T. reports grants and/or personal fees from 3ive labs, Bayer, Boehringer Ingelheim, Bristol Myers Squibb, Astra Zeneca, Novartis, Cardionomic, MagentaMed, Reprieve inc., FIRE1, W.L. Gore, Sanofi, Sequana Medical, Otsuka, Abbott, Merck, Windtree Therapeutics, Lexicon

pharmaceuticals, Precardia, Relypsa, Regeneron, BD, Edwards life sciences, and Lilly; D.G.M. and C.R.P. are co-inventors of the pending patent application "Methods and Systems for Diagnosis of Acute Interstitial Nephritis". C.R.P. is a member of the advisory board of and owns equity in RenalytixAI. He also serves as a consultant for Genfit and Novartis. In addition, J.M.T. has a patent Treatment of diuretic resistance issued to Yale and Corvidia Therapeutics Inc, a patent Methods for measuring renalase issued to Yale, and a patent Treatment of diuretic resistance pending with Reprieve inc.
