## [Peer Review File · Nature Communications]

Automated Medication-Targeted Alerts on Acute Kidney Injury Outcomes A Randomized TrialREVIEWER COMMENTS

Reviewer #1 (Remarks to the Author):

MAJOR COMMENTS

This is the latest trial from the Dr. Perry Wilson and his colleagues at Yale who have performed the most rigorous interventional studies of electronic alerts in AKI. Parallel RCT provide much stronger level of evidence compared with other designs.

The study's many strengths include the large sample size, the inclusion of several different hospitals, the fidelity to the study protocol and the original description of the study in ClinicalTrials.gov. The paper is clearly written and conforms to recommended CONSORT reporting guidelines. The actual # of outcomes 585 (23.1%) in the alert group and 639 (25.3%) in the usual care group exceeded the projected event rate of 18.2% which helped study power.

1. The choice of including PPI along with NSAID and RAS blockers is an interesting one. As the authors point out, this is not generally recommended. As a practicing nephrologist, I have to say I don't routinely recommend stopping PPI unless the urine sediment points to AIN. AIN is just much less common than ATN or "pre-renal azotemia"/cardiorenal etc hemodynamic as a cause of Cr bumps among hospitalized patients (it is more commonly a subacute decline in renal function among stable outpatients).

Yet this is the one subgroup which showed a signal and I thus think that further analysis is warranted to better understand this observation.

RE: "Within 24 hours of randomization, an MOI was discontinued in 61.1% of the alert group and 55.9% of the usual care group (relative risk 1.08, 1.04 – 1.14, p=0.0003)."

I think additional reporting of the discontinuation rates for NSAIDS, RAS blockers and PPI separately would be useful since they are quite different.

RE: "Alerts led to a 4% increase in the rate of cessation of NSAIDs (RR 1.04, 0.99 – 1.09), a 14% increase in the rate of cessation of RAASi (RR 1.14, 1.08 – 1.21), and a 26% increase in the rate of cessation of PPI (RR 1.26, 1.10 – 1.45) (Figure 2)"

I think that it will be better to present instead the differences in absolute rates of discontinuation as shown in Figure 2. The NSAID absolute discontinuation rate looks something like above 80% in the usual care group and below 80% in the alert group—likely reflecting that providers unknow NSAIDS are harmful in AKI thus will stop even without the alert.

The RAS blockers absolute discontinuation rates are lower but it appears that the absolute difference is larger.

The PPI absolute discontinuation rates are low in both arms (perhaps reflecting skepticism that PPI are causal?)—I can't quite tell but the absolute difference is about 5% (from above 20% to below 20%)? Can this realistically translate into a 4% absolute reduction in primary outcome? (31%27%?) That seems unlikely?

In a prior paper (BMJ 2021; 372 doi: <https://doi.org/10.1136/bmj.m4786>), Wilson et al did a mediation analysis. Can one be done here to see if the difference in outcome in the 2 arms are actually mediated by cessation of PPI?

2. RE: "the possibility of contamination, as providers who receive alerts may learn how better to care for AKI patients over time" Can the authors show more data about this besides just the time interaction? E.g. show Figure 2 stratified by 3-month periods to see if there are temporal trends?

3. There are some design decisions which perhaps merit more explaining of the reasons behind them E.g. why exclude those who previously had AKI requiring acute dialysis (recurrent AKI is

common and these patients may be a particularly vulnerable group which will benefit from more attention?) or exclude kidney transplant recipients (NSAIDs are also not good for them and RAS blockers are not the 1st line anti-HTN med in this population?)

4. Overall, although this is largely a "null" study, it is an important null and appropriately dampens enthusiasm for this sort of approach. I think this is an important area and there are few evidence based interventions.

Reviewer #2 (Remarks to the Author):

This pragmatic, open-label, parallel, randomized controlled trial examined whether an automated, electronic "pop-up" alert that appeared during medication order entry led to different outcomes, as measured by a primary composite endpoint of progression of AKI, dialysis, or death within 14 days or hospital discharge. Key secondary endpoints included the rate of cessation of targeted MOIs, and other subgroup analyses, all seemingly exploratory as the protocol does not specify multiplicity adjustment. The trial did not show a significant impact of the intervention on the composite primary endpoint. The authors reported other "significant" results, but this reporting should include details on the nature of these analyses, as recommended. The protocol and the manuscript are unclear regarding the definition of secondary and exploratory analyses (sometimes, these two concepts seem exchangeable).

1. (Line 176) The protocol says that two interim analyses will be done, but the design included one interim efficacy and futility analysis with 50% of enrollment (p. 19), and the final analysis. Similarly, the manuscript states, "Due to interim analyses..." (line 176). Please modify.

2. Please report the interim analysis results.

3. (Line 169) How was the 10% contamination rate between arms calculated? Is there any data on whether some providers treated patients in both arms? Can a sensitivity analysis be done after accounting for the provider? This information is crucial as this is one of the weaknesses of this trial design, i.e., contamination between arms. The authors have recognized this limitation in the discussion.

4. (Line 200) The primary endpoint/outcome results should be presented and discussed first. When showing results of other endpoints, it should be clear that no multiplicity adjustment was implemented, and therefore, all these are hypothesis-generating or exploratory analyses. For instance, the discussion section begins with a statement about the significant impact of the intervention on reducing exposure to MOI, but these findings are exploratory (Line 225). These are indeed pre-specified analyses, but the fact that the design considered no type I error rate control suggests that these are exploratory. The same comment applies to the Conclusions. If accepted in this context, the protocol-defined key secondary endpoints should be kept as such, but it should be clear that no multiplicity adjustment was implemented. A different approach should be taken for subgroup analyses (i.e., exploratory).

5. (Line 207) Additional tables like Table 1 should be presented for the MOI-exposed overlapping subgroups. For instance, the authors claimed that the alert system led to significantly improved outcomes among those exposed to PPI, but i) no details on patient characteristics by arm for this subgroup were presented, and ii) there could be confounding factors when comparing arms within this subgroup (and other MOI-defined subgroups). Were other analytic methods suitable for these situations considered when doing these subgroup analyses? It's understandable that these are exploratory analyses, but at least some details on the distribution of patient characteristics between arms should be provided, as stated before.

6. The protocol states that a retrospective examination will be performed to assess improvement in the control group before/after intervention as a measure of contamination. Please explain why the authors did not do this analysis. Again, a better understanding of contamination in this trial is crucial to interpret the results and inform future trials. Similarly, no results were reported for each institution, as stated in the protocol.

7. The statistical methods for "secondary" and "exploratory" endpoints were not specified either in the protocol or in the manuscript. For instance, the protocol says that "We will model this in exploratory analyses using an intervention-by-time of year interaction term in models of the primary outcome." However, it's not clear what model this sentence is referring to. Table 2 does not specify what methods were used to compare the length of stay and duration of AKI between

arms.

8. Safety outcomes in Table 3 were not defined in the protocol. If this is the case, please clearly state it in the manuscript.

9. Surprisingly, the trial did not report missing data. Is there missing data in this trial?

Reviewer #3 (Remarks to the Author):

This study addresses a really important question which has not previously been answered in a satisfactory way, at least for these specific medication classes. The authors evaluated three medication classes, NSAIDs, RAASi's and PPIs with differing levels of strength regarding whether the medications should be stopped, an interesting design.

The weak point of this evaluation is that it was performed within Epic which severely limits the way that decision support can be implemented. In addition, in most Epic implementations users are bombarded with warnings, which results in significant fatigue. It would be useful if the authors could offer some context about how much this is an issue in the study institutions. Regardless, a sentence about this in the limitations would be useful.

The results by class are notable and not what I expected, and thus of interest.

It would be helpful if the absolute proportions of medications that were discontinued were included in the paper--these could be added to Figure 2.

What is the clinical rationale for continuing NSAIDs in a patient with AKI? There may be one but I can't think of it--I am surprised that the discontinuation proportion did not approach 100%. I think there are better alternatives for pain control in this setting.

The biggest difference was for the PPIs which suggests that clinicians may not have thought about this category when considering what to stop.

The study could have had a stronger result if the recommendations were made more forcefully; it would be helpful to comment about that in the Discussion.

RESPONSE TO REVIEWERS' COMMENTS

The authors wish to thank the reviewers for their thoughtful and incisive comments and questions. In the following document, we have reproduced the comments and addressed the concerns point-by-point, citing specific changes to the manuscript whenever possible. We believe the manuscript has been significantly strengthened as a result of this effort and will continue to be of high impact to the *Nature Communications* readership. We wish also to reaffirm our commitment to open data sharing with regard to this manuscript, and note that the full analytic dataset is available on datadryad.com for peer-reviewers (link below) – and will be made available to the public without special conditions at the time of publication.

Reviewer #1 (Remarks to the Author):

MAJOR COMMENTS

This is the latest trial from the Dr. Perry Wilson and his colleagues at Yale who have performed the most rigorous interventional studies of electronic alerts in AKI. Parallel RCT provide much stronger level of evidence compared with other designs.

The study's many strengths include the large sample size, the inclusion of several different hospitals, the fidelity to the study protocol and the original description of the study in ClinicalTrials.gov. The paper is clearly written and conforms to recommended CONSORT reporting guidelines. The actual # of outcomes 585 (23.1%) in the alert group and 639 (25.3%) in the usual care group exceeded the projected event rate of 18.2% which helped study power.

Authors' Response: We appreciate the kind words and encouragement of this reviewer. The group continues to seek evidenced-based methods to improve the care of patients with AKI.

1. The choice of including PPI along with NSAID and RAS blockers is an interesting one. As the authors point out, this is not generally recommended. As a practicing nephrologist, I have to say I don't routinely recommend stopping PPI unless the urine sediment points to AIN. AIN is just much less common than ATN or "pre-renal azotemia"/cardiorenal etc hemodynamic as a cause of Cr bumps among hospitalize patients (it is more commonly a subacute decline in renal function among stable outpatients).

Yet this is the one subgroup which showed a signal and I thus think that further analysis is warranted to better understand this observation.

Authors' Response: We agree with this characterization of PPIs and, as the reviewer points out, deliberately targeted this class of medications based on the fact that so few physicians (even practicing nephrologists) routinely consider discontinuation. However, we believe they may contribute to more AKI than traditionally appreciated, as typically only cases of severe AKI go to kidney biopsy, where AIN can be confirmed (see PMID 25185078).

We further agree that, given the findings, further characterization of mechanism is warranted and embraced the idea of a formal mediation analysis (described and discussed below).

RE: “Within 24 hours of randomization, an MOI was discontinued in 61.1% of the alert group and 55.9% of the usual care group (relative risk 1.08, 1.04 – 1.14, p=0.0003).”

I think additional reporting of the discontinuation rates for NSAIDS, RAS blockers and PPI separately would be useful since they are quite different.

Authors' Response: We believe the reviewer may be referring to the abstract here, as the individual rates (on the relative scale at least) are reported in the results section. Given space limitations in the abstract, we felt comfortable leaving the more granular data in the body of the paper.

RE: “Alerts led to a 4% increase in the rate of cessation of NSAIDs (RR 1.04, 0.99 – 1.09), a 14% increase in the rate of cessation of RAASi (RR 1.14, 1.08 – 1.21), and a 26% increase in the rate of cessation of PPI (RR 1.26, 1.10 – 1.45) (Figure 2)”

I think that it will be better to present instead the differences in absolute rates of discontinuation as shown in Figure 2. The NSAID absolute discontinuation rate looks something like above 80% in the usual care group and below 80% in the alert group—likely reflecting that providers know NSAIDs are harmful in AKI thus will stop even without the alert.

Authors' Response: We agree that absolute discontinuation rates may convey additional information, and have added that information to the manuscript as follows:

Within 24 hours of randomization, an MOI was discontinued in 61.1% of the alert group and 55.9% of the usual care group (relative risk 1.08, 1.04 – 1.14, p=0.0003). The NSAID was discontinued in 82% of the alert group and 79% of the usual care group (RR 1.04, 0.99 – 1.09). RAASi were discontinued in 71% of the alert group and 62% of the usual care group (RR 1.14, 1.08 – 1.21). PPIs were discontinued in 22% of the alert group and 17% of the usual care group (RR 1.26, 1.10 – 1.45) (Figure 3).

The RAS blockers absolute discontinuation rates are lower but it appears that the absolute difference is larger.

Authors' Response: This is correct and a side-effect of the difference between absolute and relative scales. We hope that by providing both metrics we more completely describe the alert effect on these outcomes.

The PPI absolute discontinuation rates are low in both arms (perhaps reflecting skepticism that PPI are causal?)—I can't quite tell but the absolute difference is about 5% (from above 20% to below 20%)? Can this realistically translate into a 4% absolute reduction in primary outcome? (31%27%?) That seems unlikely?

In a prior paper (BMJ 2021; 372 doi: <https://doi.org/10.1136/bmj.m4786>), Wilson et al did a mediation analysis. Can one be done here to see if the difference in outcome in the 2 arms are actually mediated by cessation of PPI?

Authors' Response: We appreciate this and agree that the magnitude of effect on PPI discontinuation and the magnitude of effect on the clinical outcomes seem disproportionate. We have now conducted a formal mediation analysis in the PPI subgroup, which has led to the following modifications in the manuscript:

In the methods section:

We determined whether the observed clinical effects were driven by cessation of target agents by performing mediation analyses treating cessation of the MOI as a binary mediator. The product of coefficients method was used to evaluate whether MOI cessation mediates the relationship between intervention and composite outcome. Baseline covariates, including age, sex, ICU at randomization, creatinine, hemoglobin, platelet counts, the use of other MOIs and time from admission to randomization, were adjusted to control for mediator-outcome confounding, and bootstrapping was used to calculate the 95% confidence interval for proportion mediated.

In results:

Alerts may provide patient benefit beyond medication discontinuation (such as by promoting further diagnostic workup, treatment, or avoidance of nephrotoxins). As the PPI subgroup was the only one to show a significant overall effect of the alert, we performed mediation analysis to determine if the clinical effect of the alert was driven by cessation of PPI. In the unadjusted analysis without any baseline covariates (**Supplemental Table 4**), we observed that electronic health record alerts decreased the relative odds of death, dialysis, and progression of AKI by 18% (total effect on odds ratios of 0.82; 95% CI: [0.72, 0.97]), and that 10.7% of that total effect was mediated through PPI cessation (95% CI: [2.9%,44.7%]). We repeated the analysis adjusting for potential confounders, and the results remained similar.

In the discussion:

Our mediation analysis, though underpowered, suggests that there are multiple pathways of alert benefit in this population, with some benefit mediated by actual cessation of the PPI.

2. RE: “the possibility of contamination, as providers who receive alerts may learn how better to care for AKI patients over time” Can the authors show more data about this besides just the time interaction? E.g. show Figure 2 stratified by 3-month periods to see if there are temporal trends?

Authors' Response: We agree that contamination is a significant concern here, and have added a section to the results focusing on analyses of potential contamination, including the time-interaction mentioned above. In addition, we have added plots of alert effect over time to the supplemental material as suggested by the reviewer. We see no evidence of attenuation of effect over time, or based on the number of previous alerts a given provider has seen. We also provide

some historical data on the primary outcome, though we believe this is not comparable given this trial was conducted during the COVID pandemic. We have added the following language:

Assessment of Contamination

In theory, providers could “learn” to discontinue MOIs when they receive alerts, and apply that knowledge to patients randomized to usual care. To assess the extent of contamination across study arms, we conducted several analyses. First, we examined whether the effect of the alert was attenuated the longer the alert had been active in a study hospital and did not find a significant effect (interaction $p=0.51$, supplemental Figure 2A/B). Second, we examined the effect of prior provider exposure to these alerts on alert efficacy. In this analysis, we found no significant interaction between the number of alerts a provider had seen prior to a given alert and the overall efficacy of alerting ($p\text{-for-interaction}=0.48$). While we had planned to compare historical outcome rates, these are less informative than we had hoped, as this trial was conducted during the COVID pandemic, which seems to have been associated with an increased acuity of illness in this population (historical composite outcome rate for theoretically eligible patients (N=1,074) 16.8% versus 26.8% in the control group of this study).

3. There are some design decisions which perhaps merit more explaining of the reasons behind them E.g. why exclude those who previously had AKI requiring acute dialysis (recurrent AKI is common and these patients may be a particularly vulnerable group which will benefit from more attention?) or exclude kidney transplant recipients (NSAIDS are also not good for them and RAS blockers are not the 1st line anti-HTN med in this population?)

Authors' Response: We appreciate that design decisions in clinical trials can seem opaque or even capricious but have done our best to better discuss the reasons for these particular exclusions in the revised manuscript. The section on participants now includes the following:

Patients were excluded if their initial creatinine after admission was greater than or equal to 4.0 mg/dL or if they had received dialysis within the year prior to meeting the AKI definition (as these may be patients with end-stage kidney disease cared for outside of our health system), if they had been admitted to a hospice service or had an active “comfort measures only” order, if they had a diagnosis code consistent with end-stage kidney disease, or if they had a kidney transplant within the 6 months prior to randomization (as transplant recipients at our institution receive specialist nephrology care during their inpatient stay, regardless of AKI).

4. Overall, although this is largely a "null" study, it is an important null and appropriately dampens enthusiasm for this sort of approach. I think this is an important area and there are few evidence based interventions.

Authors' Response: We are grateful the reviewer recognizes the value of negative trials, when appropriately conducted. We look forward to ongoing work to optimize care of the patient with AKI and agree that one-size-fits-all alerting (or even more tailored drug-specific alerting) may not be adequate to move the needle on outcomes.

Reviewer #2 (Remarks to the Author):

This pragmatic, open-label, parallel, randomized controlled trial examined whether an automated, electronic “pop-up” alert that appeared during medication order entry led to different outcomes, as measured by a primary composite endpoint of progression of AKI, dialysis, or death within 14 days or hospital discharge. Key secondary endpoints included the rate of cessation of targeted MOIs, and other subgroup analyses, all seemingly exploratory as the protocol does not specify multiplicity adjustment. The trial did not show a significant impact of the intervention on the composite primary endpoint. The authors reported other “significant” results, but this reporting should include details on the nature of these analyses, as recommended. The protocol and the manuscript are unclear regarding the definition of secondary and exploratory analyses (sometimes, these two concepts seem exchangeable).

Authors' Response: We thank the reviewer for their suggestions on how we can improve clarity with regards to the endpoints assessed in this trial. We have updated the manuscript to more clearly indicate primary, secondary, exploratory, and safety outcomes as outlined in response to the following questions.

1. (Line 176) The protocol says that two interim analyses will be done, but the design included one interim efficacy and futility analysis with 50% of enrollment (p. 19), and the final analysis. Similarly, the manuscript states, “Due to interim analyses...” (line 176). Please modify.

Authors' Response: Thank you for calling this to our attention. Our original protocol called for two analyses, one at 50% enrollment in the teaching hospitals, followed by expansion to the non-teaching hospitals (assuming no safety signals existed) and then a second interim analysis at 50% enrollment in the non-teaching hospitals. This design was chosen given our signal of harm from our first AKI alert trial which was restricted to the two non-teaching hospitals. However, despite a lack of concerning safety signals at the first interim analysis (50% of the teaching hospitals), the IRB did not grant our request to expand the study to the non-teaching hospitals. As such, we modified the protocol to recruit entirely from the teaching hospitals and thus no second interim analysis was performed. We have clarified this in the manuscript as follows:

The original protocol called for two interim analyses, one at 50% recruitment across four teaching hospitals, and one at 50% recruitment at two non-teaching hospitals, as our prior trial showed a potential signal of harm at the non-teaching hospitals. However, despite no harm signal at the initial interim analysis, the IRB did not grant our request to extend the study into the two non-teaching hospitals. We thus completed recruitment entirely in the four teaching hospitals and completed only one interim analysis.

2. Please report the interim analysis results.

Authors' Response: We now report the interim analysis results as follows:

On the basis of a prespecified interim analysis (N=1,980), which found that the primary outcome occurred in 255 (24.7%) of individuals in the alert group and 254 (26.8%) of those in the usual care group (p=0.30), the external DSMB recommended the trial proceed to full recruitment.

3. (Line 169) How was the 10% contamination rate between arms calculated? Is there any data on whether some providers treated patients in both arms? Can a sensitivity analysis be done after accounting for the provider? This information is crucial as this is one of the weaknesses of this trial design, i.e., contamination between arms. The authors have recognized this limitation in the discussion.

Authors' Response: We agree that contamination is one of, and perhaps the main limitation to this study design. We cite several simulation studies that suggest that increasing sample size is a reasonable tool to account for bias (towards the null) in the face of contamination, where clustering is not feasible (as is the case for our study), but acknowledge that 10% is somewhat arbitrary. As described in our response to the reviewer 1, we have added a section to the results dedicated to evaluating contamination in several ways reproduced (here as well), which now (at the reviewers suggestion, also includes an analysis of whether alert effect is modified by the number of prior alerts the provider had seen). We see no evidence of contamination in these analyses.

Assessment of Contamination

In theory, providers could “learn” to discontinue MOIs when they receive alerts, and apply that knowledge to patients randomized to usual care. To assess the extent of contamination across study arms, we conducted several analyses. First, we examined whether the effect of the alert was attenuated the longer the alert had been active in a study hospital and did not find a significant effect (interaction $p=0.51$, supplemental Figure 2A/B). Second, we examined the effect of prior provider exposure to these alerts on alert efficacy. In this analysis, we found no significant interaction between the number of alerts a provider had seen prior to a given alert and the overall efficacy of alerting ($p\text{-for-interaction}=0.48$). While we had planned to compare historical outcome rates, these are less informative than we had hoped, as this trial was conducted during the COVID pandemic, which seems to have been associated with an increased acuity of illness in this population (historical composite outcome rate for theoretically eligible patients ($N=1,074$) 16.8% versus 26.8% in the control group of this study).

4. (Line 200) The primary endpoint/outcome results should be presented and discussed first. When showing results of other endpoints, it should be clear that no multiplicity adjustment was implemented, and therefore, all these are hypothesis-generating or exploratory analyses. For instance, the discussion section begins with a statement about the significant impact of the intervention on reducing exposure to MOI, but these findings are exploratory (Line 225). These are indeed pre-specified analyses, but the fact that the design considered no type I error rate control suggests that these are exploratory. The same comment applies to the Conclusions. If accepted in this context, the protocol-defined key secondary endpoints should be kept as such, but it should be clear that no multiplicity adjustment was implemented. A different approach should be taken for subgroup analyses (i.e., exploratory).

Authors' Response: We appreciate this thoughtful advice. As the secondary outcomes reported were pre-specified, we have removed the reference to those outcomes as “exploratory”. We have clarified that there was no adjustment for multiplicity in the methods. Subgroup analyses were also pre-specified, but we appreciate the lack of power in this setting to limit our ability to make inferences here. As such, we now describe the subgroup analyses as exploratory. We have also updated the methods to read:

Secondary outcomes are evaluated at a 2-sided p-value threshold of 0.05 without multiplicity adjustment and should be considered hypothesis generating. Subgroup analyses, while pre-specified, should be considered exploratory.

5. (Line 207) Additional tables like Table 1 should be presented for the MOI-exposed overlapping subgroups. For instance, the authors claimed that the alert system led to significantly improved outcomes among those exposed to PPI, but i) no details on patient characteristics by arm for this subgroup were presented, and ii) there could be confounding factors when comparing arms within this subgroup (and other MOI-defined subgroups). Were other analytic methods suitable for these situations considered when doing these subgroup analyses? It's understandable that these are exploratory analyses, but at least some details on the distribution of patient characteristics between arms should be provided, as stated before.

Authors' Response: While, ideally, randomization should balance baseline covariates even within subgroups, we appreciate that chance imbalances can affect results. As such, we have added Supplemental tables to our supplement file essentially recreating Table 1 for each MOI (supplemental tables 3, 4, and 5).

6. The protocol states that a retrospective examination will be performed to assess improvement in the control group before/after intervention as a measure of contamination. Please explain why

the authors did not do this analysis. Again, a better understanding of contamination in this trial is crucial to interpret the results and inform future trials. Similarly, no results were reported for each institution, as stated in the protocol.

Authors' Response: As discussed above, and in response to reviewer 1, we have added a section to the results focusing on various methods to assess contamination. We have added historical outcome data, though we find it less informative as the historical data is pre-COVID and it certainly seems that the acuity of patients is substantially higher in this cohort compared to a historical cohort who would have been theoretically eligible for this trial. These analyses are added to the section on contamination. We also appreciate that effects may differ across institutions, as seen in our prior alert trial. We have thus added the institution-specific effect sizes to our subgroup analysis figure.

7. The statistical methods for “secondary” and “exploratory” endpoints were not specified either in the protocol or in the manuscript. For instance, the protocol says that “We will model this in exploratory analyses using an intervention-by-time of year interaction term in models of the primary outcome.” However, it’s not clear what model this sentence is referring to. Table 2 does not specify what methods were used to compare the length of stay and duration of AKI between arms.

Authors' Response: We appreciate the need to provide more granularity with regard to the statistical analyses of these outcomes. We have amended the methods section as follows:

We compared categorical variables across the treatment groups with the use of the Cochran-Mantel-Haenszel Chi-squared test stratified by the four study hospitals. We assessed continuous outcomes using the Van Elteren test, accounting for stratification by study hospital.

In addition, we have updated the footer of Table 2 to now read:

Table 2: Secondary outcomes. Unless otherwise specified, all outcomes are evaluated within 14 days of randomization or hospital discharge and expressed as count, percentage. AKI = Acute Kidney Injury. Difference in duration of Aki and LOS were assessed via the Van Elteren test (accounting for clustering within hospitals), and were not significantly different [duration of AKI (p=0.14), length of stay (p=0.38)].

8. Safety outcomes in Table 3 were not defined in the protocol. If this is the case, please clearly state it in the manuscript.

Authors' Response: We agree that the safety definitions were not fully operationalized in the protocol, however we did specify the particular safety concerns in the “risks” section (end of page 8 through page 9). There, we note the monitoring plan for these outcomes. We feel our description of the safety metrics is adequate to reflect that these theoretical concerns did not occur during the study.

9. Surprisingly, the trial did not report missing data. Is there missing data in this trial?

Authors' Response: We appreciate the importance of characterizing missing data in all studies. In this case, there is no missing data with regard to the primary outcome (due in large part to the fact that the outcome was defined based upon outcomes that occur during hospitalization). However, we do note a small amount of missing data in terms of baseline characteristics. We have updated the supplemental material to reflect this and replicate that data here:

Supplemental Table 8 – Missing Data

Covariate	Missing in (number, %)
Baseline anion gap	29 (0.6%)
Baseline serum bicarbonate	24 (0.5%)
Baseline BUN	26 (0.6%)
Baseline chloride	26 (0.5%)
Baseline hemoglobin	36 (0.7%)
Baseline platelet count	44 (0.9%)
Baseline potassium	29 (0.6%)
Baseline sodium	26 (0.5%)
Baseline white blood cell count	43 (0.9%)

Supplemental Table 8 – Missing Data. All outcome data was complete for all enrolled participants. Counts of missing baseline covariates are reported here.

Additionally, we have modified the manuscript to now read:

Data was collected electronically and confirmed by two chart reviewers on a random sample of alert and control patients (N=100). There was no missing data on outcomes, and missing data overall was scarce (supplemental Table 8).

Reviewer #3 (Remarks to the Author):

This study addresses a really important question which has not previously been answered in a satisfactory way, at least for these specific medication classes. The authors evaluated three medication classes, NSAIDs, RAASi's and PPIs with differing levels of strength regarding whether the medications should be stopped, an interesting design.

Authors' Response: We appreciate the reviewer's enthusiasm for the topic!

The weak point of this evaluation is that it was performed within Epic which severely limits the way that decision support can be implemented. In addition, in most Epic implementations users are bombarded with warnings, which results in significant fatigue. It would be useful if the authors could offer some context about how much this is an issue in the study institutions. Regardless, a sentence about this in the limitations would be useful.

Authors' Response: We certainly agree that we are limited in our approach based on the tools in use within our health system. We further agree that alert fatigue is a substantial concern in studies like this, and have added the following to the discussion:

Additionally, our alerts did not exist in a vacuum, but in competition with all the other clinical alerts and distractions that are enabled by the EHR. Alert fatigue may diminish alert performance.

The results by class are notable and not what I expected, and thus of interest.

Authors' Response: To us as well. Although it was always a goal of this study to look at the effects by class, as (internally at least) we have often thought of this as 3 trials in one.

It would be helpful if the absolute proportions of medications that were discontinued were included in the paper--these could be added to Figure 2.

Authors' Response: Per the above reviewers' comment, we have added this to the text of the manuscript. We have not amended Figure 2 in order to not clutter the image (and duplicate data that now appears in text).

What is the clinical rationale for continuing NSAIDs in a patient with AKI? There may be one but I can't think of it--I am surprised that the discontinuation proportion did not approach 100%. I think there are better alternatives for pain control in this setting.

Authors' Response: Discontinuation rates were obviously quite high, but we agree that 100% would be reasonable. That said, even in the context of this study it is possible that providers simply did not notice mild AKI, or recognize the presence of an NSAID.

The biggest difference was for the PPIs which suggests that clinicians may not have thought about this category when considering what to stop.

Authors' Response: We believe this is probably true. Baseline rates of PPI cessation are rather low, and clinically this is simply not a drug most clinicians consider when they are thinking about “nephrotoxins” as classically defined.

The study could have had a stronger result if the recommendations were made more forcefully; it would be helpful to comment about that in the Discussion.

Authors' Response: This is an interesting point, but we worry it might be too complicated for this particular manuscript. In fact, the “tone” of the alert was the subject of substantial back-and-forth with the IRB during the study design phase. There was concern from the IRB that too forceful language would lead to reflexive / unthinking actions on the part of the provider, which would expose patients to more-than-minimal risk. (As a study operating under a waiver of informed consent, the risk could not be more than minimal to study participants). As such, we had to ensure that the language allowed for providers to use their clinical discretion. This might not be an issue were alerts implemented in a non-research setting, of course.

REVIEWERS' COMMENTS

Reviewer #1 (Remarks to the Author):

I appreciate the opportunity to review this manuscript again. The authors have done a very good job addressing the points I raised earlier. There is one remaining point I hope they can clarify.

For these new results: "electronic health record alerts decreased the relative odds of death, dialysis, and progression of AKI by 18% (total effect on odds ratios of 0.82; 95% CI: [0.72, 0.97]), and that 10.7% of that total effect was mediated through PPI cessation (95% CI: [2.9%,44.7%])."

So is it 10.7% of 18% or about 1.8%? (or is it 10.7% out of 18% which is more than half of the effect size?)

Reviewer #2 (Remarks to the Author):

Authors addressed concerns satisfactorily. Except for the comment below, I have no additional comments or concerns.

- The mediation analysis must state that it was a non-protocol specified analysis.

Reviewer #3 (Remarks to the Author):

Regarding the point below, I still ask the authors to make a brief comment about this in the discussion. The IRB's perspective is probably incorrect in the greater scheme of things, and is a good example of an IRB being too intrusive from my perspective. But it is fine for the authors to state that had a dialogue with the IRB about the strength of the recommendations and that they respected the IRB's recommendations.

This probably had a significant impact on the likelihood that the recommendations would be accepted, and I do not think it is a fine point.

**

The study could have had a stronger result if the recommendations were made more forcefully; it would be helpful to comment about that in the Discussion.

Authors' Response: This is an interesting point, but we worry it might be too complicated for this particular manuscript. In fact, the "tone" of the alert was the subject of substantial back-andforth with the IRB during the study design phase. There was concern from the IRB that too forceful language would lead to reflexive / unthinking actions on the part of the provider, which would expose patients to more-than-minimal risk. (As a study operating under a waiver of informed consent, the risk could not be more than minimal to study participants). As such, we had to ensure that the language allowed for providers to use their clinical discretion. This might not be an issue were alerts implemented in a non-research setting, of course.

We sincerely appreciate the thoughtful comments and questions from the reviewers and are happy to provide further detail.

Herein, we transcribe the reviewers' questions and respond. Where there have been substantive changes in the manuscript text, we include those sections within a text box. These are also, of course, reflected in the changes-tracked version of our manuscript. Thank you again for the consideration.

REVIEWERS' COMMENTS

Reviewer #1 (Remarks to the Author):

I appreciate the opportunity to review this manuscript again. The authors have done a very good job addressing the points I raised earlier. There is one remaining point I hope they can clarify.

For these new results: "electronic health record alerts decreased the relative odds of death, dialysis, and progression of AKI by 18% (total effect on odds ratios of 0.82; 95% CI: [0.72, 0.97]), and that 10.7% of that total effect was mediated through PPI cessation (95% CI: [2.9%,44.7%])."

So is it 10.7% of 18% or about 1.8%? (or is it 10.7% out of 18% which is more than half of the effect size?)

Response to reviewer: We appreciate the opportunity to clarify this. This is 10.7% of the 18% - so 10% of the overall effect size of 18% (or an isolated effect of 1.8%). Of course, mediation analyses are somewhat underpowered (note the CI ranging from 2.9 to 44.7%). Regardless, it is clear that the act of stopping PPI may not be the main driver of benefit in this population. To clarify that to readers we have modified the discussion as follows:

Our mediation analysis, though underpowered, suggests that there are multiple pathways of alert benefit in this population, with some, but likely not the majority, benefit mediated by actual cessation of the PPI.

Reviewer #2 (Remarks to the Author):

Authors addressed concerns satisfactorily. Except for the comment below, I have no additional comments or concerns.

- The mediation analysis must state that it was a non-protocol specified analysis.

Response to reviewer: We appreciate this concern. However, the protocol states "Finally, should alerting show a clinical benefit, we will perform a mediation analysis to determine what fraction of that benefit (if any) can be attributed to medication cessation." We feel that this indicates that

consideration of a mediation analysis preceded analysis of the data.

Reviewer #3 (Remarks to the Author):

Regarding the point below, I still ask the authors to make a brief comment about this in the discussion. The IRB's perspective is probably incorrect in the greater scheme of things, and is a good example of an IRB being too intrusive from my perspective. But it is fine for the authors to state that had a dialogue with the IRB about the strength of the recommendations and that they respected the IRB's recommendations.

This probably had a significant impact on the likelihood that the recommendations would be accepted, and I do not think it is a fine point.

Response to Reviewers: Thank you for this comment. We agree that we can more forcefully discuss the limitations placed on this study by the IRB. As an aside, we are very interested in the ethical implications of IRBs limiting what can be effectively studied in this space when interventions can be implemented without hard evidence. This is one of those cases where, had the alert been implemented broadly under the guise of “quality improvement” – very forceful recommendations could be made. However, to actually *study* alerts like this, we are more limited. That disconnect will need to be resolved should implementation science studies like this continue to move forward.

We have revised the relevant portion of the discussion as follows:

Moreover, in order to meet requirements set out by the IRB, the language of the alert was limited in how stringently it could recommend medication cessation; more forceful wording could have increased cessation rates, and, potentially, clinical effectiveness.